# Surface-Potential-Based Compact Modeling of p-GaN Gate HEMTs

**DOI:** 10.3390/mi12020199

**Published:** 2021-02-15

**Authors:** Jie Wang, Zhanfei Chen, Shuzhen You, Benoit Bakeroot, Jun Liu, Stefaan Decoutere

**Affiliations:** 1The Key Laboratory for RF Circuit and Systems of Ministry of Education, Hangzhou Dianzi University, Hangzhou 310012, China; wangjie@hdu.edu.cn (J.W.); ljun77@hdu.edu.cn (J.L.); 2Interuniversity Microelectronics Centre (IMEC), Kapeldreef 75, B-3001 Leuven, Belgium; shuzhen.you@imec.be (S.Y.); stefaan.decoutere@imec.be (S.D.); 3CMST, IMEC, Ghent University, Technologiepark 126, B-9052 Ghent, Belgium; Benoit.Bakeroot@UGent.be

**Keywords:** p-GaN gate high-electron mobility transistors, compact model, physics-based models, surface potential

## Abstract

We propose a surface potential (SP)-based compact model of p-GaN gate high electron mobility transistors (HEMTs) which solves the Poisson equation. The model includes all possible charges in the GaN channel layer, including the unintended Mg doping density caused by out-diffusion. The SP equation and its analytical approximate solution provide a high degree of accuracy for the SP calculation, from which the closed-form I–V equations are derived. The proposed model uses physical parameters only and is implemented in Verilog-A code.

## 1. Introduction

GaN based high electron mobility transistors (HEMTs), which are typically normally-on devices, draw significant attention for power switching applications due to the combined merits of a high OFF-state voltage and low ON-resistance R_ON_ originating from the high electron mobility and large breakdown electric field in these materials [1]. Yet the market demands for normally-off or enhancement mode (E-mode) devices. One of the approaches to obtain E-mode transistors is to use a p-GaN gate that lifts up the conduction band at the channel surface in equilibrium [2] and results in normally-off operation [3] by depleting the 2-D electron gas (2DEG) in the channel [4]. Today, the p-GaN gate devices were the first commercialized E-mode devices with high and robust threshold voltage V_th_, high breakdown voltage, and low dynamic R_ON_ increase [5].

Effective circuit design using these devices requires fast and accurate models that can describe the behavior of the device under different working conditions. Previous work includes models such as the Curtice model [6], the Statz model [7], and the Angelov–Chalmers model [8], which are mostly empirical curve-fitting models that might not work well for large signal operation. DynaFET [9] is an innovative nonlinear model based on a non-linear network analyzer (NVNA) and an artificial neural network (ANN). These models hardly represent the physical GaN HEMT behavior, for they lack any predictive capability and are strictly limited to the test system. To overcome these difficulties, an exact, complete, and simple physics-based model is required. Two industrial standard models were recently selected by the CMC (Compact Model Council). First, there is the ASM model [10], a physics-based surface-potential (SP) model able to capture terminal characteristics of GaN HEMTs by solving the Schroedinger–Poisson coupled equations. Secondly, unlike SP-based models, the MIT Virtual Source GaN (MVSG) model [11] formulates mobile charge densities through a threshold voltage term. The physics-based MVSG model adopts a different interpretation of the carrier velocity by using an empirical saturation function for GaN HEMTs. Recently, a charge-based HEMT model was developed by EPFL [12], starting from a physics-based model for regular silicon FETs and was given new physical quantities typical for HEMTs. However, the above models lack the physics of the typical gate structure in the p-GaN gate HEMTs, consisting of a “back-to-back” diode configuration [13].

Previously, we have developed a SP-based model for Schottky gate GaN HEMTs [14] which solved the problem of elaborate computations of the surface potential by using an approximate analytical solution. The goal of this work is to extend our model of the p-GaN HEMTs, by considering within the Poisson equation all possible charges induced: the polarization charges at the AlGaN/GaN and p-GaN/AlGaN interfaces, the active Mg doping in the AlGaN barrier and the GaN channel caused by out-diffusion [4], and holes. Our SP-based model is physics-based and precisely captures all operation regions in the Direct Current (DC) behavior. The model accuracy was demonstrated by the excellent agreement between the model and experimental data including Current–Voltage (I-V) and Capacitance–Voltage (C-V) results. The p-GaN gate HEMT model was implemented in Verilog-A, and the main characteristics of the device can be reproduced.

This brief is arranged as follows: In Section II, we present the device structure. Section III is divided in two subsections: (1) we describe the drain current (I_D_) and gate current (I_G_) models and (2) the gate, drain, and source charges are calculated. The model’s implementation, parameter extraction, and experimental validation are discussed in Section IV. Finally, we conclude this paper.

## 2. Device Structure 

The p-GaN gate HEMTs were fabricated in 200 mm pilot line using an Au-free process flow. Details of the process flow are described in [4]. Figure 1 shows a cross-section transmission electron microscope (TEM) picture of the gate area of a realized p-GaN gate HEMT. Table 1 is the main material properties in p-GaN gate HEMT.

## 3. Surface Potential (SP)-Based Model

### 3.1. Current Calculation

First of all, it is important to concentrate on modeling the potential distribution along the channel of the p-GaN gate HEMTs. In order to obtain the electrostatic potential φ distribution in the p-GaN gate HEMTs, the 1D Poisson equation is solved in the GaN channel layer, in which the carrier density is determined by the polarization charges and the p-GaN gate at equilibrium [2]. Additionally, the p-GaN HEMTs feature unintended Mg doping density in the channel caused by out-diffusion, and thus the Poisson equation, including all possible charges within the device, must be considered. Since both the AlGaN and GaN layers are usually undoped, the depletion charge can be neglected. For completeness, we included hole charge density in the Poisson equation:(1)$$\frac{d^{2}\varphi }{{\mathrm{dx}}^{2}}+\frac{d^{2}\varphi }{{\mathrm{dy}}^{2}}=\frac{q}{\varepsilon_{\mathrm{GaN}}}\left(N_{\mathrm{Mg}}+n_{\mathrm{GaN}}-p_{\mathrm{GaN}}\right)$$
where ε_GaN_ is the permittivity of GaN, N_Mg_ is the unintended Mg doping in the channel caused by out-diffusion, φ is the surface potential, x is the direction along channel width, y is the direction perpendicular to the channel, n_GaN_ and p_GaN_ are the electrons’ and holes’ charge densities in the GaN channel layer, respectively.

The n_GaN_ and p_GaN_ are written as:(2)$$n_{\mathrm{GaN}}=n_{i}\cdot \exp\left(\left(-\Phi_{F}+\varphi -V_{C}\right)/V_{T}\right)$$
(3)$$p_{\mathrm{GaN}}=n_{i}\cdot \exp{\left(\left(\Phi_{F}-\varphi \right)/V_{T}\right)}_{\;}$$

The n_GaN_ and p_GaN_ can be represented in terms of the intrinsic carrier concentration n_i_, the Fermi potential Φ_F_ (Φ_F_ = V_T_[ln(N/n_i_)+2^−1.5^(N/n_i_)]; N is the carrier concentration), the thermal voltage V_T_ (V_T_ = kT/q), and the voltage V_C_ applied between the channel and the substrate.

If we assume the gradual channel approximation that ignores the lateral field gradient in the Poisson equation, that is, ignore the y dependence of (x; y) in Equation (1), then we get:(4)$$\frac{d^{2}\varphi }{{\mathrm{dx}}^{2}}=\frac{q}{\varepsilon_{\mathrm{GaN}}}\left[n_{i}\exp\left(\frac{\varphi -\Phi_{F}-V_{C}}{V_{T}}\right)-n_{i}\exp\left(\frac{\Phi_{F}-\varphi }{V_{T}}\right)+N_{\mathrm{Mg}}\right]$$

Defining a Gaussian surface from the GaN channel down to the neutral substrate gives the potential at x: φ(x) = 0 for x → ∞ and dφ(x)/dx = 0 for x → ∞. Imposing these boundary conditions gives, at x = 0, φ(x = 0) = φ_GaN_ at the surface, where φ_GaN_ is the surface potential.

With the two-order integral from φ = 0 to φ = φ_GaN_ of Equation (4), we can get the Surface-Potential Equation (SPE) shown in Equation (5), and this is the basis for our model.
(5)$$V_{G}-V_{0}-\varphi_{\mathrm{GaN}}=\gamma \sqrt{\begin{array}{c} \frac{\varphi_{\mathrm{GaN}}}{V_{T}}N_{\mathrm{Mg}}+n_{i}\exp\left(\frac{-\Phi_{F}-V_{C}}{V_{T}}\right)\left[\exp\left(\frac{\varphi_{\mathrm{GaN}}}{V_{T}}\right)-1\right]\\ +n_{i}\exp\left(\frac{\Phi_{F}}{V_{T}}\right)\left[\exp\left(-\frac{\varphi_{\mathrm{GaN}}}{V_{T}}\right)-1\right] \end{array}}$$

In Equation (5), V_G_ is the gate voltage; γ is the body factor; V_0_ is a fitting parameter such that the surface potential φ_GaN_ equals zero when V_G_ = V_0_. We use an advanced non-iterative algorithm [15] to solve Equation (5) in a self-consistent way and obtain the potential distribution along the surface.

The surface potential φ_pGaN_ at the p-GaN/AlGaN layer interface is calculated with the same methodology [15].
(6)$$\frac{d^{2}\varphi^{\prime }}{{\mathrm{dx}}^{2}}+\frac{d^{2}\varphi^{\prime }}{{\mathrm{dy}}^{2}}=\frac{q}{\varepsilon_{\mathrm{pGaN}}}\left(N_{\mathrm{Mg}}^{\prime }+n_{\mathrm{pGaN}}-p_{\mathrm{pGaN}}\right)$$
where ε_pGaN_ is the permittivity of AlGaN; $N_{\mathrm{Mg}}^{\prime }$ is the Mg doping in the p-GaN layer; n_pGaN_ and p_pGaN_ are the electrons’ and holes’ charge densities in the p-GaN layer, respectively.

The drain current is the most important device parameter of the HEMT device. The model uses drift-diffusion transport to calculate the drain current and adopts the channel current equation of the thin layer charge model [16] as:(7)$$I_{D}=-\mu W\left(q_{i}\frac{{d\varphi }_{\mathrm{GaN}}}{\mathrm{dy}}-V_{T}\frac{{\mathrm{dq}}_{i}}{\mathrm{dy}}\right)$$
where I_D_ is the drain current, q_i_ is the charge density in the GaN channel, μ is the electron mobility, and W is the gate width.

The p-GaN gate HEMTs, fabricated at IMEC, feature a TiN gate metal on top of the p-GaN layer in order to form a Schottky contact on the p-GaN to 2DEG gate input diode, to reduce the leakage current. Therefore, the gate leakage follows a path along this “back-to-back diode” structure [2].

According to the “back-to-back” diodes structure in the p-GaN gate HEMTs, the Schottky junction is reverse biased while the p-GaN/AlGaN/GaN junction is forward biased for higher gate voltages. The gate current can be calculated from the thermionic emission (TE) current over the AlGaN/GaN heterojunction with the current continuity condition [17]:(8)$$J_{\mathrm{TE}}=\mathrm{Area}\cdot J_{\mathrm{TE}0}T^{2}\left[\exp\left(\frac{\varphi_{\mathrm{pGaN}}-\varphi_{\mathrm{GaN}}}{n_{\mathrm{TE}}V_{T}}\right)-1\right]$$
where Area is the area of the gate, J_TE0_ is the current density of heterojunction which depends on the AlGaN/GaN barrier height, T is the temperature, and n_TE_ is the ideality factor for heterojunction.

At V_G_ < 0 V, the Schottky junction is forward biased, while the p-GaN/AlGaN/GaN junction is reverse biased, and the gate current is determined by a reverse leakage current along the edges of the gate and across the heterojunction. For this reverse leakage current, the hopping transport (HT) [18] model has been developed. The gate current empirically obeys the following equation:(9)$$J_{\mathrm{HT}}=\mathrm{Peri}\cdot J_{\mathrm{HT}0}\left[\exp\left(\frac{\varphi_{\mathrm{pGaN}}}{n_{\mathrm{HT}}V_{T}}\right)-1\right]\exp\left(-T^{-\alpha }\right)$$
where Peri is the perimeter of the gate, J_HT0_ is the reverse saturation current density, n_HT_ is the ideality factor, and α is the temperature coefficient [19]. 

### 3.2. Charge Calculation

The gate, drain, and source charges can be calculated by integrating the channel charge (*q_ch_*) along the channel following the Ward–Dutton charge partitioning [20] as:(10)$$\{\begin{array}{c} Q_{G}=W\int_{0}^{L}q_{ch}dx\\ Q_{D}=W\int_{0}^{L}q_{ch}\frac{x}{L}dx\\ Q_{S}=W\int_{0}^{L}q_{ch}\left(1-\frac{x}{L}\right)dx \end{array}$$ where *L* is the channel length, *W* is the channel width and *x* is the position along the channel. 

The integration in Equation (10) can be solved using the Symmetric Linearization Method [21] and neglecting the effect of velocity saturation on the charge.

The complete p-GaN gate HEMT structure contains several field plates (FPs), and therefore, attention is given to the accurate modeling of the FPs within the device. Given the field plate configuration depicted in Figure 2, the 2DEG is present below the FP at equilibrium. These field plate regions can be regarded as additional transistors in series with different threshold voltages. Hence, the field plate regions can be regarded as additional transistors in series with different operating voltages. The C–V behavior of a p-GaN gate HEMT is of utmost importance for the high-frequency performance and switching characteristics of these high-power devices, and this is intrinsically related to the FP configuration. The equivalent circuit of capacitances is shown in Figure 2. The C–V characteristics can be modeled considering the capacitors with varying insulator thickness.

## 4. Results and Discussion

We have validated the proposed SP-based compact model using I–V and C–V measurement data of a typical 200 V p-GaN gate high-power HEMT device with a total device width W of 36 mm. The model is in good agreement with the measured currents, especially the gate current, and capacitances, as shown in Figure 3, Figure 4, Figure 5, Figure 6 and Figure 7. For transconductance characteristics in Figure 3b, the effect of high electric field is lack in this model, which will be considered in the follow-up work to improve the accuracy. 

The simulated transfer and output characteristics match well with the measurement results at 300 K in Figure 3 and Figure 4, sweeping V_G_ from −1 V to 6 V with V_D_ from 0.1 V to 2.2 V in I_D_-V_G_, and V_D_ from −1 V to 10 V with V_G_ from 2 V to 7 V in I_D_-V_D_. The OFF-state capacitances C_GS_, C_GD_, and C_DS_ at V_G_ = 0 V with V_D_ from 0 V to 200 V, are nicely modeled in Figure 5, capturing the impact of the field plates that modulate the 2DEG distribution at high V_D_. The developed compact SP-based model successfully reproduces the measured current and capacitance characteristics.

The temperature dependence characteristics are also modelled and shown in Figure 6.

The Figure 7 shows an excellent agreement between the model and the measurement for both branches of V_G_ ≥ 0 V and V_G_ < 0 V at different temperatures of 300 and 425 K.

## 5. Conclusions

The proposed SP-based compact model for p-GaN gate HEMTs is based on a surface-potential analytical approximate solution, which considers all possible induced charges at the AlGaN/GaN interface. This model gets the complete potential distribution along the surface by solving the Poisson equation. The developed compact model is physical as it includes the main GaN HEMT features in the solution of the Poisson equation. Good fit to the measurement results was obtained over a large range of gate and drain voltages. It was verified that this model can reproduce all observed current and capacitance characteristics of p-GaN gate HEMTs automatically.

## Figures and Tables

**Figure 1 micromachines-12-00199-f001:**
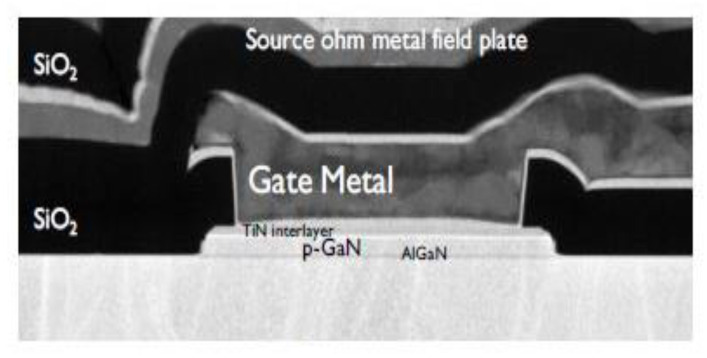
Transmission electron microscope(TEM) picture of the Schottky metal/p-GaN gate AlGaN/GaN high electron mobility transistors (HEMT).

**Figure 2 micromachines-12-00199-f002:**
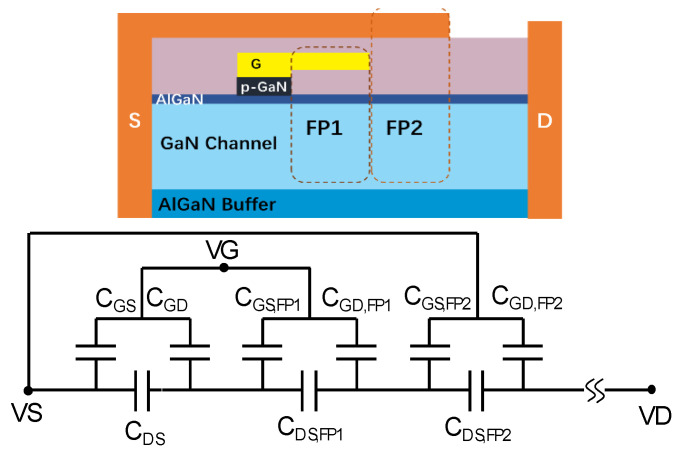
The equivalent circuit of the capacitance of field plates (FPs) of a p-GaN gate HEMT.

**Figure 3 micromachines-12-00199-f003:**
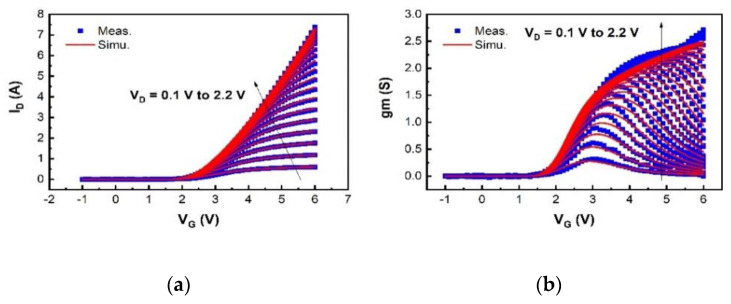
Measurement data (dots) and simulation data (line) at 300 K with V_D_ from 0.1 to 2.2 V, step = 0.1 V. (**a**) Transfer characteristics; (**b**) transconductance characteristics.

**Figure 4 micromachines-12-00199-f004:**
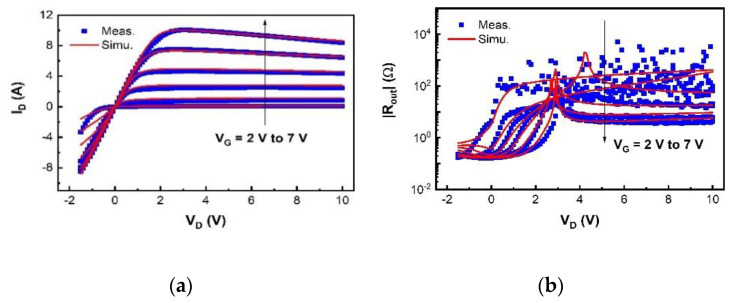
Measurement data (dots) and simulation data (line) at 300 K with V_G_ from 2 to 7 V, step = 1 V. (**a**) Output characteristics; (**b**) absolute values of output resistance characteristics.

**Figure 5 micromachines-12-00199-f005:**
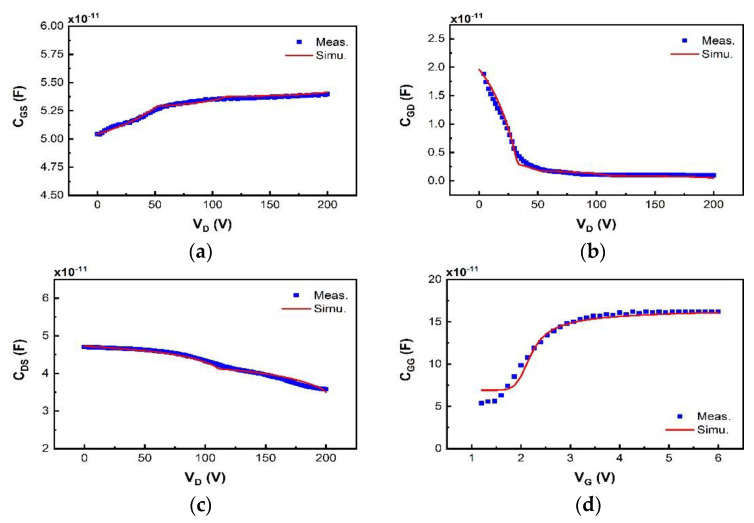
The C–V characteristics of the surface potential (SP) model (line) are compared with measurement data (dot) as a function of input voltage: (**a**) C_DS_ at different V_D_; (**b**) C_GD_ at different V_D_; (**c**) C_GS_ at different V_D_; (**d**) C_GG_ (= C_GS_ + C_GD_) at different V_G_.

**Figure 6 micromachines-12-00199-f006:**
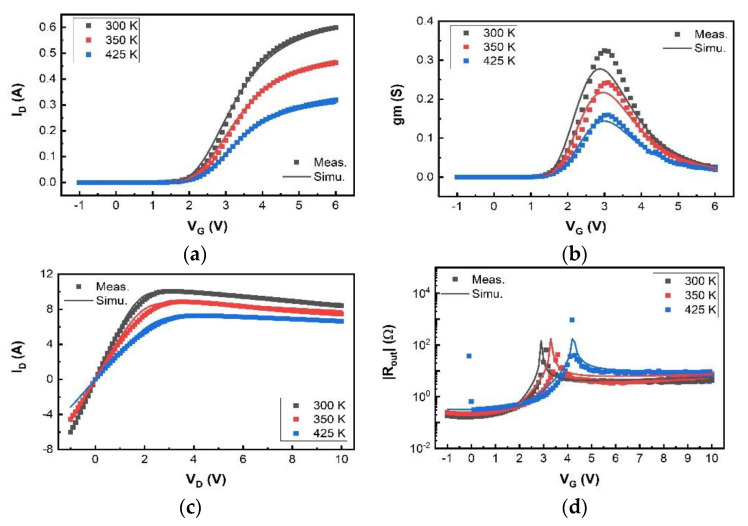
Measurement data (dots) and simulation data (line) at different temperatures. (**a**) Transfer characteristics with V_D_ = 0.1 V; (**b**) transconductance resistance characteristics with V_D_ = 0.1 V; (**c**) output characteristics with V_G_ = 7 V; (**d**) output resistance characteristics with V_G_ = 7 V.

**Figure 7 micromachines-12-00199-f007:**
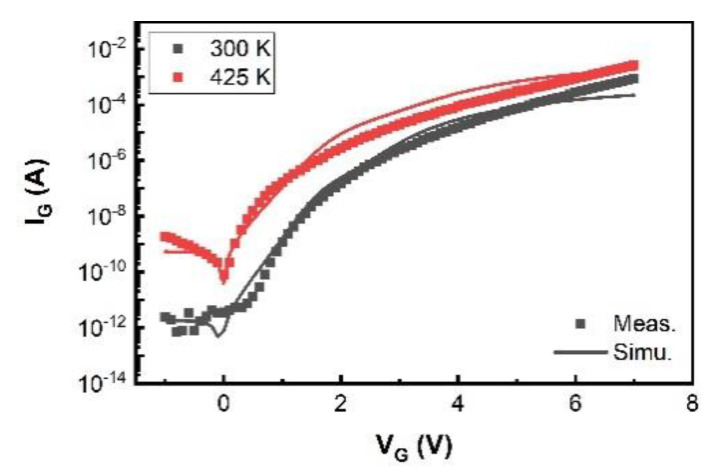
Measurement (dots) and simulation (lines) data of I_G_ at 300 and 425 K with V_DS_ = 0.1 V.

**Table 1 micromachines-12-00199-t001:** Main material properties in p-GaN gate HEMT.

Parameter	Physical Meaning	p-GaN HEMT
*N_Mg_*	Mg density	3~5 × 10^17^ cm^−3^
*E_g-GaN_*	GaN Band gap	3.4 eV
*∆E_C_*	The Difference between the Conductor Bottom and Fermi Level	0.518 eV
*N_C_*	GaN Conduction band density	2.02 × 10^18^ cm^−3^
*N_V_*	GaN Valence band density	9.08 × 10^18^ cm^−3^
*ε_GaN_*	GaN dielectric constant	9.4
*ε_AlGaN_*	AlGaN dielectric constant	8.94

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
