# Peer review of "Surface-Potential-Based Compact Modeling of p-GaN Gate HEMTs"

_micromachines, 2021, doi:10.3390/mi12020199_

Round 1

Reviewer 1 Report

Thank you for this good paper that provides interesting results. You will find below some comments :

  • In this paper you should add the epi-structure used for the fabrication of the device.

  • Did your simulator give you the right Rsheet and Ns measured on the epi-structure?

  • Can you add also in the text the width of the measured device.

  • The tunneling phenomenon is happening during the operation of the device in several locations below the gate. Did you consider this phenomenon in your simulator?

  • In figure3 we observe an increase of Gm at Vgs over 5.5V for high Vds on measurement data not on the simulation ones. Can you please explain the origin of this increase of Gm and why it’s not present in your simulation results.

  • Can you plot the Ids(Vgs) curves in log scale to see if your model is fitting well the subthreshold behavior of the device.

  • In figure4 b) you should specify that you are presenting the absolute value of Rout.  

  • In figure5, can you zoom in on CGD plot in the flat zone to verify that your field plates are correctly modeled.

  • Please identify CGG.

  • Figure6, for Ids(Vds) measurement you choose Vgs=8V that seems to be very high, can you explain the raison for that?

Author Response

Dear Reviewer,

Thank you very much for your review of my article, and here's my response to your questions. Please see the attachment. The content involved has been corrected in the article. Thank you.

Reviewer 2 Report

Dear Mr. Chen,

Thank you for assigning me 1099581 to review. The authors model and measure a high electron mobility transistor, obtaining reasonable results. The approach seems likely to have industrial applications. There are a large number of papers about HEMT models already.

I recommend major revision and resubmission. My main concern is that the work relies heavily on methods from Ref. 16, which I could not easily access and does not appear to be peer reviewed. Perhaps this reference material needs to be published first. Detailed critiques are listed below.

1. l. 67 "imec" is unclear. It should not be necessary to refer to the author's affiliations to figure out what "imec" is.
2. Fig. 1(b) looks like a screenshot from some other document which might need attribution.
3. Fig. 1 is incompletely and inconsistently labeled.
4. Eq. (1) It is better to define all the symbols, even if clear from context.
5. Eq. (2-3) are missing a brief explanation.
6. l. 91-93 could have clearer grammar.
7. Eq. 5 The bounds of integration were not explicitly stated. Maybe this is implied by the previous paragraph.
8. Eq. 10 What is W? Differences from Ref. 21 Eq. 12 need explanation.
9. Fig. 2 should be more clearly related to Fig. 1.
10. Fig. 3(b) does not show "good agreement" as claimed. In addition, the legibility is poor.
11. Fig. 6(a) and (c) The lines are illegible.
12. l. 168 "Explicitly" does not seem appropriate unless there is an equation giving the solution. It is contradicted on lines 97-98.
13. l. 168 "more physical" compared to what? Including which features?
14. There should be a table of assumed material properties with references.
15. The Verilog-A code was not provided and therefore cannot be reviewed.

Author Response

(The authors gave the same response as above.)

Round 2

Reviewer 2 Report

Dear Editor Chen,

The authors have not addressed comments 4, 5, 6, 8, 10, and 14 in the text.  Therefore my advice is unchanged.  I do not wish to review this work a third time.

Comment 7:  Bounds of integration with respect to x should be bounds of x.

Fig. 2 still has substantial room for improvement.

The purpose of response letter pages 3-8 is unclear.